# Age Weakens the Other-Race Effect among Han Subjects in Recognizing Own- and Other-Ethnicity Faces

**DOI:** 10.3390/bs13080675

**Published:** 2023-08-11

**Authors:** Jialin Ma, Rui Zhang, Yongxin Li

**Affiliations:** Facuty of Education, School of Psychology, Henan University, Kaifeng 475000, China; zhangrui2000@henu.edu.cn (R.Z.); liyongxin@henu.edu.cn (Y.L.)

**Keywords:** age, Tibetan, Han, face recognition, other-race effect, eye movement

## Abstract

The development and change in the other-race effect (ORE) in different age groups have always been a focus of researchers. Previous studies have mainly focused on the influence of maturity of life (from infancy to early adulthood) on the ORE, while few researchers have explored the ORE in older people. Therefore, this study used behavioral and eye movement techniques to explore the influence of age on the ORE and the visual scanning pattern of Han subjects recognizing own- and other-ethnicity faces. All participants were asked to complete a study-recognition task for faces, and the behavioral results showed that the ORE of elderly Han subjects was significantly lower than that of young Han subjects. The results of eye movement showed that there were significant differences in the visual scanning pattern of young subjects in recognizing the faces of individuals of their own ethnicity and other ethnicities, which were mainly reflected in the differences in looking at the nose and mouth, while the differences were reduced in the elderly subjects. The elderly subjects used similar scanning patterns to recognize the own- and other-ethnicity faces. This indicates that as age increases, the ORE of older people in recognizing faces of those from different ethnic groups becomes weaker, and elderly subjects have more similar visual scanning patterns in recognizing faces of their own and other ethnicities.

## 1. Introduction

The phenomenon in which subjects recognize own-race faces more accurately than other-race faces is the other-race effect (ORE) [1]. OREs exist not only in face recognition between different races but also between different ethnicities in the same race (race refers to a population with certain common genetic characteristics in physical form) [2,3]. For example, Uighur and Han subjects also experience an ORE when recognizing the faces of their own and other ethnicities [2]. Both Tibetan and Han subjects belong to the East Asian race; however, there are OREs when Tibetan and Han subjects recognize faces of their own and other ethnicities [3].

In recent years, eye movement technology has been widely used to explore the differences in scanning patterns caused by the ORE because it can directly reflect the visual processing characteristics of faces [4,5,6]. Researchers hoped to explore the reasons for the ORE by comparing the cross-racial visual scanning patterns of faces. Studies of eye movement have found that subjects may use different scanning patterns when recognizing own- and other-race faces [7,8,9,10]. Some studies have found that the differences in processing strategies [4,11] and face configurations were the main reasons for the differences in scanning patterns of other-race face recognition, and these differences may be the reason for the ORE [12,13]. First, subjects of different races use different processing strategies when scanning faces [4,11]. For example, to explore the visual processing characteristics of other-race face recognition, some studies have divided faces into three areas of interest (AOIs): the eyes, nose and mouth areas. The results showed that Caucasian subjects focused more on the eyes of the face than other facial features when they recognized their own-race and East Asian faces [12,13]. In contrast, East Asian subjects focused more on the central area of the face (eyes and nose) when they recognized their own-race and Caucasian faces than on other facial features [4,9,11,14]. Researchers believe that Caucasian subjects can obtain more features from the eyes compared with other facial features, so they pay more attention to the eyes [12]. East Asian subjects adopt the holistic processing strategy for processing whole faces from the center of the face to the surrounding area [15,16]. The upper bridge of the nose between the eyes is the central area of the East Asian face [3,17]. Therefore, compared with other facial features, East Asian subjects focus more on the eyes and nose, which belong to the central area and reflect the different eye movement processing patterns of East and West subjects to a certain extent when processing own- and other-race faces.

Second, some studies have found that differences in face configurations lead to differences in the visual processing of other-race faces [4,12,13]. For example, McDonnell et al. (2014) believed that the differences in face configurations were the main reasons for the differences in the visual scanning patterns of Caucasian and African subjects [18]. McDonnell et al. divided faces into three AOIs, the eyes, nose and mouth, and asked Caucasian subjects to scan their own-race and African faces. The results showed that Caucasian subjects fixated more on the upper half, such as the area containing the eyes, of own-race faces than on African faces, while the fixation on the lower half, such as the area containing the nose, of own-race faces was fixated on significantly less than the lower half of African faces. McDonnell et al. believed that the reason Caucasian subjects pay more attention to the nose area of African faces was that compared with Caucasian faces, African faces have longer and wider noses, which could attract more attention from subjects. Ma et al.’s (under review) research also supports this viewpoint. The research divided faces into three AOIs: the eyes, nose and mouth. Tibetan and Han subjects were required to complete a learning-recognition task for own- and other-ethnicity (ethnicity refers to a group of people different from other groups based on history, culture and language) faces. The results showed that there was an ORE in the recognition of faces of their own and other ethnicities by Tibetan and Han subjects; Tibetan and Han subjects focused more on the eyes and mouth areas when recognizing Han faces and focused more on the eyes and nose areas when recognizing Tibetan faces. Ma et al. believed that the reason for the differences in visual scanning patterns between Tibetan and Han subjects was the differences in the configuration of Tibetan and Han faces [13,19]. Tibetan and Han subjects had an East Asian cultural background, and both adopted the scanning pattern of processing from the center of the face to the surrounding area [3,9]. Han faces are mostly round or oval, with the center point between the eyes, while Tibetan faces have a longer nose and mouth, and their face shape is longer than that of Han faces, with the center point on the upper bridge of the nose (see Figure 1 for details) [3]. Therefore, when Tibetan and Han subjects recognize faces of their own and other ethnicities, there are differences in their scanning patterns due to different facial configurations.

It is notable that the processing face of subjects is often regulated by age [20], so the influence of age on the ORE has also been widely studied by researchers. Previous studies have mainly focused on exploring the reasons for the differences in the ORE between young children and individuals in early adulthood [5,21] and found that the differences in experience with face-to-face contact [22] and visual processing styles [6] were the main reasons for the differences in the ORE between the two groups. First, the ORE is closely related to face-to-face experiences. A study found that there was no lateralization in a newborn’s gaze on own- and other-race faces [23], but the ORE was observed in infants 3–6 months old [24,25,26,27]. In early adulthood, the ORE becomes more stable [22,28,29]. Researchers believe that the development and change in the ORE from the newborn stage to early adulthood were related to the change in the subjects’ experience with own- and other-race faces [22,28,30,31,32,33]. In the early stage of infancy, subjects have no face-to-face contact experience with own- and other-race faces, and there is no difference in their recognition. However, when subjects gradually accumulate experience with their own-race faces, the recognition of their own-race faces becomes increasingly specialized. Therefore, compared with other-race face recognition, own-race face recognition is better, and the ORE appears. Second, children and early adults have different visual scanning patterns for own- and other-race faces [6]. On the one hand, with increasing age, children of different races show an increasing number of scanning patterns that are consistent with their own-race cultural background. For example, for Caucasian subjects, their fixation of the eye area of Caucasian faces increases with age [6], and scanning patterns are consistent with the routine of maintaining eye contact in social communication in Caucasian culture [34]. This result is also supported by other studies [27,34], while research from East Asian subjects found that Asian infants aged 4 to 9 months focused more on the central area of Asian faces [5], which was more similar to the visual scanning patterns of early adult subjects [8,35,36]. Therefore, with increasing age, their fixation patterns on faces are increasingly close to those of their own-race adults. Of course, before adulthood, this difference causes a variation in the ORE between the two age groups.

In summary, from infancy to early adulthood, the main reasons for the differences in the ORE are the changes in the subjects’ experience with faces of their own and other races and the development and changes in the visual processing pattern, which are affected by culture. In addition, age is an important factor affecting the ORE, and a comparative study of the ORE at different ages could help to clarify the mechanism underlying the development of the ORE. However, the above studies only focused on the development and changes in the ORE from young children to individuals in early adulthood. Existing studies have mainly focused on the differences in scanning patterns between early adulthood and elderly adulthood when individuals recognize faces of individuals from their own ethnic group [37]. For example, compared to young people, elderly subjects tend to adopt a holistic mode, in which most eye gaze is concentrated in the center of the face, especially elderly people with lower cognitive states [38]. For facial features, both young and elderly people consider the eyes and nose as the main focus areas [37], whereas elderly people tend to pay more attention to their mouths than young people [39]. However, focusing on only how young and elderly people process the faces of individuals from their own ethnic groups is not sufficient, and thus far, whether the ORE exists in elderly adulthood is unclear. Therefore, this study selected Tibetan and Han faces as materials, took young and elderly Han people as subjects, and used ethology and eye movement techniques to explore the impact of aging on the ORE and other-race face scanning patterns of Han subjects.

First, the difference between elderly and young Han subjects’ experience with faces of other races affects the magnitude of the ORE. The contact hypothesis posits that increased contact between members of different social groups can help reduce prejudice and discrimination between groups. Therefore, based on this hypothesis, we believe that the appearance of an ORE is related to the fact that individuals have less contact experience with other-race faces than with own-race faces [22,28,40]. In recent decades, the rapid development of China’s economy has been inseparable from the economic and trade exchanges between ethnic groups, and trade and cultural exchanges between ethnic groups are becoming increasingly frequent. Under this background, elderly people have many opportunities to contact faces of individuals of different ethnicities in advanced age, while young people are mostly at the stage of schooling or just entering society, with less contact experience with different ethnic faces. Therefore, compared with young people, elderly people have more experience with faces of individuals of different ethnicities. Thus, we propose the first hypothesis: compared with young people, elderly people have a rich contact experience with faces of individuals of other ethnicities that may reduce the ORE of elderly Han subjects in recognizing own- and other-ethnicity faces.

Additionally, there may be differences between elderly and young Han subjects in the visual processing of faces of their own and other ethnicities. According to research [3], affected by the cultural background of East Asia, East Asian subjects adopt the visual processing pattern of scanning from the center to the face to the surrounding area when recognizing faces of their own and other ethnicities. This phenomenon was also proven by a previous study [41,42]. The study found that the East Asian Han subjects used the scanning pattern from the center of the face to the surrounding area when recognizing the Tibetan and Han faces, but the two scanning patterns were different due to the variations in the configuration of the Tibetan and Han faces. Han subjects focused more on the eyes and mouth area when recognizing Han faces and more on the eyes and nose area when recognizing Tibetan faces, so the difference was mainly reflected in the fixation on the nose and mouth. Both the young and elderly Han subjects used the scanning pattern of processing from the center to the face to the surrounding area to recognize the faces of their own and other ethnicities. Therefore, young and elderly Han subjects focus more on the eyes in the center of Han faces and the eyes and nose in the center of Tibetan faces.

In addition, research has found that elderly people are better at the vertical processing of faces than young people; that is, the processing of the nose and mouth of elderly persons is significantly greater than that of young persons [20]. However, there was no significant difference between elderly and young persons in the horizontal configuration of faces; that is, there was no difference between elderly and young persons in eye fixation [20]. It could be inferred that compared with young subjects, elderly subjects increase their fixation on the nose and mouth of Tibetan and Han faces, while the study found that the differences in subjects’ scanning patterns when recognizing faces of their own and other ethnicities were mainly observed in the nose and mouth area [3,42]. The increase in the elderly subjects’ fixation on the nose and mouth of own- and other-ethnicity faces will probably reduce or eliminate the differences in the visual scanning patterns of the cross-ethnic faces. In summary, we propose a second hypothesis: there may be differences in the scanning patterns of own- and other-ethnicity faces among young Han subjects, mainly reflected in the gaze on the nose and mouth, while the scanning patterns of own- and other-ethnicity faces among elderly subjects are likely to have no such differences or fewer differences.

## 2. Method

### 2.1. Participants

Seventy-seven Han subjects (all Han Chinese participants came from concentrated areas of Han, and the majority of Han people in daily life are less likely to come into contact with other ethnic groups), including 35 young subjects (21.30 ± 3.27 years old, 16 males) and 42 elderly subjects (61.52 ± 6.3 years old, 24 males), were randomly recruited. All subjects had no history of mental illness and no insomnia, anxiety or other symptoms in the recent month. All subjects were right-handed, had normal vision or corrected-to-normal vision, and signed informed consent documents. After the experiment was performed, all subjects were rewarded with a small monetary payment in exchange for their time.

### 2.2. Experimental Materials

A total of 192 neutral facial expression images used in the experiment were selected from the “facial-expression database of Chinese Han, Hui and Tibetan people” [39]. There were 96 facial images in the learning stage, including 48 Tibetan faces (24 male faces) and 48 Han faces (24 male faces). In addition, 96 new facial images were presented in the recognition stage, including 48 Tibetan faces (24 male faces) and 48 Han faces (24 male faces). Photoshop CS6 was used for all faces to remove hair, ears and other irrelevant factors. Images had a resolution of 640 × 480 pixels and were converted to grayscale images. The study divided faces into three AOIs, the eyes, nose and mouth, as shown in Figure 2. The area of each face was approximately 8 cm^2^. The face was divided into three AOIs (eyes, nose and mouth); the area of the eyes was approximately 1.7 cm^2^, the area of the nose was approximately 1.8 cm^2^ and the area of the mouth was approximately 1.6 cm^2^.

### 2.3. Equipment

Experiment Builder 1.4.0 software was used during the experiment, and eye movement data were recorded by the EyeLink^®^ 1000 Plus eye tracker(manufacturer: SR Research Ltd., Ottawa, ON, Canada). The sample rate was 1000 Hz, the line-of-sight error was accurate within 5°, the screen resolution was 1024 × 768 pixels, and the distance between the subjects and the display screen was 65 cm. The screen angle of view was 15.5° × 11.7° and the facial image angle of view was 9.85° × 7.5°.

### 2.4. Procedure

The experiment adopted the “study-recognition” paradigm. In the learning stage, first, instructions were presented to subjects, who were required to carefully study the 96 randomly presented face pictures, and subjects were told that their learning performance would be tested later. After the subjects carefully read the instructions, they pressed any key on the keyboard to start learning. On the screen, a cross-ethnic face image was presented for 5 s, followed by the presentation of a fixation point for 1 s [37]; then, the next facial expression image was presented. This process was repeated for a total of 96 trials. To ensure the accuracy of the data, a single-point calibration was added to each trial. The positions of single-point calibration randomly appeared at the top, middle, bottom, left and right of the screen. Eyelink 1000 Plus was used to record the eye movement data (fixation counts and fixation duration) of the subjects in the learning stage [16,38].

In the recognition stage, the subjects were required to recognize 192 faces (96 of them were learned faces and 96 were new faces). If the face was a learned face, participants were to press the F key, and if it was a new face, participants were to press the J key. The subjects needed to press the key quickly while ensuring accuracy. In the experiment, the subjects’ recognition accuracy was recorded. This part of the data was collected by the subjects’ dictation, and the experimenter recorded it.

### 2.5. Research Design and Data Analysis

A total of 77 effective eye movement data points were collected (35 young subjects, 42 elderly subjects), and SPSS 25.0 was used to analyze the behavioral and eye movement data. We ran a 2 age of subjects (young group, elderly group) × 2 ethnicity of faces (Han face, Tibetan face) repeated-measures ANOVA to evaluate the accuracy at the recognition stage and a 2 age of subjects (young group, elderly group) × 2 ethnicity of faces (Han face, Tibetan face) × 3 AOI (eyes, nose and mouth) repeated-measures ANOVA to evaluate the fixation counts and fixation duration. The fixation counts (the total counts of subjects’ gazes at a certain AOI) and the fixation duration (the total duration of subjects’ gazes at a certain AOI) were the main indicators used to measure the allocation of attention resources in the AOI in the face of subjects [12]; these indicators are widely used in face stimulation eye-tracking research.

## 3. Results

### 3.1. Behavioral Data

#### Accuracy

Repeated-measures ANOVA with 2 ages of subjects (young group, elderly group) × 2 ethnicities of faces (Han face, Tibetan face) was conducted to assess recognition accuracy, and the study used the Bonferroni correction method to correct the results. The results showed that the main effect of face ethnicity was significant, *F*(1,75) = 37.44, *p* < 0.001, η_p_^2^ = 0.33. The recognition accuracy of Han faces (*M* = 0.60, *SD* = 0.06) was significantly higher than that of Tibetan faces (*M* = 0.57, *SD* = 0.07). The main effect of subjects’ age was not significant, *F*(1,75) = 2.70, *p* = 0.11, η_p_^2^ = 0.04. The interaction between the ethnicity of the face and the age of the subjects was significant, *F*(1,75) = 6.42, *p* < 0.05, η_p_^2^ = 0.08. Through simple effect analysis, it was found that in the young group, the recognition accuracy of Han faces was significantly higher than that of Tibetan faces, *F*(1,75) = 34.31, *p* < 0.001, η_p_^2^ = 0.31. The recognition accuracy of Han faces in the elderly group was significantly higher than that of Tibetan faces, *F*(1,75) = 8.98, *p* < 0.01, η_p_^2^ = 0.11, which proved that the young and elderly group subjects experienced an ORE when recognizing faces of their own and other ethnicities. The recognition accuracies of Tibetan and Han faces by subjects in the young and elderly groups are presented in Table 1, and the interactions are shown in Figure 3.

To further explore the influence of age on the ORE size, the study conducted an independent sample T test on the ORE (other-race effect = recognition accuracy of own-race faces—recognition accuracy of other-race faces) of the young and elderly group subjects. The results showed that the ORE of the elderly group was significantly smaller than that of the young group, t = 2.53, *p* < 0.05, d = 0.37.

### 3.2. Eye Movement Data

#### 3.2.1. Fixation Duration for Facial AOIs

Repeated-measures ANOVAs with 2 ages of subjects (young group, elderly group) × 2 ethnicity of face (Han face, Tibetan face) × 3 AOI (eyes, nose and mouth) were conducted to analyze fixation duration on AOIs. The study used the Bonferroni correction method to correct the results. The results showed that the main effect of facial AOI was significant, *F*(2,74) = 31.67, *p* < 0.001, η_p_^2^ = 0.30. The fixation duration on the eyes (*M* = 1564.86, *SD* = 534.11) was significantly longer than that on the nose (*p* < 0.001, *M* = 1115.28, *SD* = 476.19) and mouth (*p* < 0.001, *M* = 1166.73, *SD* = 532.21). The main effect of face ethnicity was significant, *F*(1,75) = 12.00, *p* < 0.001, η_p_^2^ = 0.14. The subjects’ fixation duration on Han faces (*M* = 1325.98, *SD* = 672.41) was significantly longer than that on Tibetan faces (*M* = 1238.60, *SD* = 574.63). The main effect of age was significant, *F*(1,75) = 6.15, *p* < 0.05, η_p_^2^ = 0.08, and the fixation duration of the young group (*M* = 1316.96, *SD* = 643.27) was significantly longer than that of the elderly group (*M* = 1247.63, *SD* = 501.44).

The interaction between AOI and face ethnicity was significant, *F*(2,150) = 14.27, *p* < 0.001, η_p_^2^ = 0.16. Through simple effect analysis, it was found that the subjects’ fixation durations on the eyes of Tibetan faces was significantly longer than those on the nose (*p* < 0.001) and mouth (*p* < 0.001), and the fixation durations on the nose were significantly longer than those on the mouth (*p* < 0.001), *F*(2,74) = 18.15, *p* < 0.001, η_p_^2^ = 0.33. The subjects’ fixation durations on the eyes (*p* < 0.05) and mouth (*p* < 0.001) of the Han faces were significantly longer than those on the nose, *F*(2,74) = 24.42, *p* < 0.001, η_p_^2^ = 0.40. The fixation durations for the mouths of Han faces were significantly greater than those for Tibetan faces, *F*(1,75) = 22.03, *p* < 0.001, η_p_^2^ = 0.23. The fixation durations for the nose of Tibetan faces were significantly greater than those for Han faces, *F*(1,75) = 7.89, *p* < 0.01, η_p_^2^ = 0.10, and the fixation durations for the eyes of Han and Tibetan faces were not significantly different, *F*(1,75) = 1.02, *p* = 0.32, η_p_^2^ = 0.01. The means and standard deviations of the fixation durations of subjects in the AOIs of Tibetan and Han faces are presented in Table 2.

The interaction effect of the ethnicity of the face and participant age was not significant, *F*(1,75) = 0.41, *p* = 0.52, η_p_^2^ = 0.005. The interaction between subjects’ age and AOI was also not significant, *F*(2,150)= 1.12, *p* = 0.33, η_p_^2^ = 0.015. The interaction of face ethnicity, age and AOI was significant, *F*(2,150) = 5.97, *p* < 0.01, η_p_^2^ = 0.07. Through simple effect analysis, it was found that when young subjects recognized Tibetan faces, the fixation duration on the eyes was significantly longer than that on the nose (*p* < 0.05) and mouth (*p* < 0.001), and the fixation duration on the nose was significantly longer than that on the mouth (*p* < 0.001), *F*(2,74) = 15.27, *p* < 0.001, η_p_^2^ = 0.29. When young subjects recognized Han faces, the fixation duration on the eyes (*p* < 0.001) and mouth (*p* < 0.001) was significantly longer than that on the nose, *F*(2,74) = 20.77, *p* < 0.001, η_p_^2^ = 0.36. When elderly subjects recognized Tibetan faces, the fixation duration on the eyes was significantly longer than those on the mouth (*p* < 0.01) and nose (*p* < 0.01), *F*(2,74) = 5.45, *p* < 0.01, η_p_^2^ = 0.13. When elderly subjects recognized Han faces, the fixation duration on the eyes was significantly longer than those on the mouth (*p* < 0.01) and nose (*p* < 0.01), *F*(2,74) = 6.09, *p* < 0.01, η_p_^2^ = 0.14. When the young subjects recognized Han and Tibetan faces, the fixation durations for the mouth of Han faces were significantly greater than those for Tibetan faces, *F*(1,75) = 21.47, *p* < 0.001, η_p_^2^ = 0.22; the fixation durations for the nose of Tibetan faces were significantly greater than those for Han faces, *F*(1,75) = 14.93, *p* < 0.001, η_p_^2^ = 0.17; and the fixation durations for the eyes of Han and Tibetan faces were not significantly different, *F*(1,75) = 0.13, *p* = 0.72, η_p_^2^ = 0.002. When the elderly subjects recognized Han and Tibetan faces, the fixation durations for the mouth (*F*(1,75) = 15.22, *p* < 0.001, η_p_^2^ = 0.17) of Han faces were significantly greater than those for Tibetan faces, and the fixation durations for the eyes (*F*(1,75) = 1.21, *p* = 0.27, η_p_^2^ = 0.016), nose (*F*(1,75) = 0.004, *p* = 0.95, η_p_^2^ = 0.000) and mouth (*F*(1,75) = 3.56, *p* = 0.06, η_p_^2^ = 0.045) of Han and Tibetan faces were not significantly different. The means and standard deviations of fixation durations of subjects in different age groups for the AOIs of Tibetan and Han faces are presented in Table 3. The interactions are shown in Figure 4a–c.

#### 3.2.2. Fixation Counts for Facial AOIs

Repeated-measures ANOVAs with 2 ages of subjects (young group, elderly group) × 2 ethnicities of face (Han face, Tibetan face) × 3 AOIs (eyes, nose and mouth) were conducted to analyze fixation counts of facial AOIs. The study used the Bonferroni correction method to correct the results. The results showed that the main effect of facial AOIs was significant, *F*(2,74) = 34.36, *p <* 0.001, η_p_^2^ = 0.31. The fixation counts of the eyes (*M* = 5.46, *SD* = 2.12) were significantly greater than those of the nose (*p* < 0.001, *M* = 3.91, *SD* = 4.07) and mouth (*p* < 0.001, *M* = 4.07, *SD* = 2.23). The main effect of face ethnicity was significant, *F*(1,75) = 14.38, *p* < 0.001, η_p_^2^ = 0.14. The subjects’ fixation counts for Han faces (*M* = 1238.60, *SD* = 574.63) were significantly greater than those for Tibetan faces (*M* = 1325.98, *SD* = 672.41). The main effect of age was significant, *F*(1,75) = 16.59, *p* < 0.001, η^2^ = 0.18. The fixation counts of the young group (*M* = 4.65, *SD* = 1.90) were significantly greater than those of the elderly group (*M* = 4.25, *SD* = 1.87).

The interaction between AOI and face ethnicity was significant, *F*(2,150) = 35.34, *p* < 0.001, η_p_^2^ = 0.32. Through simple effect analysis, it was found that the subjects’ fixation counts for the eyes of Tibetan faces were significantly greater than those for the nose (*p* < 0.001) and mouth (*p* < 0.001), and the fixation counts for the nose were significantly greater than those for the mouth (*p* < 0.001), *F*(2,74) = 34.06, *p* < 0.001, η_p_^2^ = 0.48. The fixation counts for the eyes (*p* < 0.001) and mouth (*p* < 0.001) of the Han faces were significantly greater than those for the nose, *F*(2,74) = 37.95, *p* < 0.001, η_p_^2^ = 0.51. The fixation counts for the mouths (*F*(1,75) = 21.94, *p* < 0.001, η_p_^2^ = 0.23) of Han faces were significantly greater than those for Tibetan faces, *F*(1,75) = 56.76, *p* < 0.001, η_p_^2^ = 0.43; the fixation counts for the noses of Tibetan faces were significantly greater than those for Han faces, *F*(1,75) = 21.94, *p* < 0.001, η_p_^2^ = 0.23; and the fixation counts for the eyes of Han and Tibetan faces were not significantly different, *F*(1,75) = 0.71, *p* = 0.40, η_p_^2^ = 0.01. The means and standard deviations of the fixation times of subjects in the AOIs of Tibetan and Han faces are presented in Table 2.

The interaction effect of face ethnicity and age was not significant, *F*(1,75) = 0.05, *p* = 0.82, η_p_^2^ = 0.001. The interaction between subjects’ age and AOI was not significant, *F*(2,150) = 2.71, *p* = 0.07, η_p_^2^ = 0.04.

The interaction of face ethnicity, age and AOI was significant, *F*(2,150) = 5.59, *p* < 0.01, η_p_^2^ = 0.07. After a simple effect analysis, it was found that when young subjects recognized Tibetan faces, the fixation counts of the eyes were significantly greater than those of the nose (*p* < 0.01) and mouth (*p* < 0.001), and the fixation counts of the nose were significantly greater than those of the mouth (*p* < 0.001), *F*(2,74) = 30.52, *p* < 0.001, η_p_^2^ = 0.45. When the young subjects recognized Han faces, the fixation counts of the eyes (*p* < 0.001) and mouth (*p* < 0.001) were significantly greater than those of the nose, *F*(2,74) = 28.63, *p* < 0.001, η_p_^2^ = 0.44. When the elderly subjects recognized Tibetan faces, the fixation counts of the eyes were significantly greater than those of the mouth (*p* < 0.001) and nose (*p* < 0.05), *F*(2,74) = 5.45, *p* < 0.01, η_p_^2^ = 0.13. When the elderly subjects recognized Han faces, the fixation counts of the eyes (*p* < 0.001) and mouth (*p* < 0.001) were significantly greater than those of the nose, *F*(2,74) = 10.73, *p <* 0.001, η_p_^2^ = 0.23. When the young subjects recognized Han and Tibetan faces, the fixation counts for the mouths of Han faces were significantly greater than those for Tibetan faces, *F*(1,75) = 44.08, *p* < 0.001, η_p_^2^ = 0.37; the fixation counts for the noses of Tibetan faces were significantly greater than those for Han faces, *F*(1,75) = 25.45, *p* < 0.001, η_p_^2^ = 0.25; and the fixation counts for the eyes of Han and Tibetan faces were not significantly different, *F*(1,75) = 0.14, *p* = 0.72, η_p_^2^ = 0.002. When the elderly subjects recognized Han and Tibetan faces, the fixation counts for the mouths (*F*(1,75) = 15.22, *p* < 0.001, η_p_^2^ = 0.17) of Han faces were significantly greater than those for Tibetan faces, and the fixation counts for the eyes (*F*(1,75) = 0.70, *p* = 0.41, η_p_^2^ = 0.009) and noses (*F*(1,75) = 2.10, *p* = 0.16, η_p_^2^ = 0.03) of Han and Tibetan faces were not significantly different. The means and standard deviations of fixation counts of subjects in different age groups for the AOIs of Tibetan and Han faces are presented in Table 4. The interactions are shown in Figure 5a–c.

## 4. Discussion

This study used behavioral and eye movement techniques to explore the ORE and visual scanning patterns of elderly and young Han subjects in recognizing own- and other-ethnicity faces. The results showed that the ORE occurred when elderly and young Han subjects recognized the faces of their own race and those of Tibetans. Moreover, the ORE of the elderly subjects was significantly smaller than that of the young subjects, which showed that aging reduces the ORE and supported our first hypothesis. Evidence from eye movement also supported this viewpoint. When recognizing Han faces, young subjects gazed most at the eyes and mouth areas, followed by the nose. When recognizing Tibetan faces, young subjects gazed most at the eyes, followed by the nose and then the mouth. The differences in the visual scanning patterns of young subjects in recognizing Tibetan and Han faces were mainly reflected in the eyes and nose. However, the elderly subjects gazed most on the eyes of Tibetan and Han faces, followed by the nose and mouth, and there was no significant difference in the fixation duration on the AOIs of the Tibetan and Han faces, which supported the second hypothesis. These findings showed that elderly subjects used similar scanning patterns to recognize own- and other-ethnicity faces, and age had a regulatory effect on the recognition of cross-ethnic faces and their visual browsing patterns.

### 4.1. ORE and Scanning Patterns of Cross-Ethnic Face Recognition

The ORE occurs when Han subjects recognize Han and Tibetan faces and adopt different scanning patterns for Tibetan and Han faces. The behavioral data showed that the memory performance of the Han subjects in recognizing Han faces was significantly higher than that in recognizing Tibetan faces; that is, there was an ORE, which was also proven by the eye movement evidence that the Han subjects’ fixation (fixation counts and duration) on their own-race faces was significantly greater than that on Tibetan faces. The results proved that the Han subjects demonstrated an ORE when recognizing faces of their own and other ethnicities, which was consistent with previous research results [3]. We believe that the following reasons may have caused this result. First, the lack of encounters with other-race faces may be the main reason for the occurrence of the ORE [28,30]. Although both Tibetan and Han people are East Asians, the subjects selected in this study were Tibetan and Han people who belong to compact communities of ethnicities. There were few opportunities to encounter other-race faces; therefore, subjects lacked experience in processing other-race faces, which led to the ORE. Second, different face configurations also lead to an ORE. Some studies found that differences in face configuration also affect other-race face recognition [3,16]. The significant difference in the face configuration between Tibetan and Han people [3] led to Han subjects not accurately recognizing Tibetan faces, which led to the appearance of the ORE.

To further analyze the reasons for the ORE in Han subjects’ recognition of Tibetan and Han faces, we conducted a comparative analysis of the visual scanning patterns of Han subjects who scanned Han and Tibetan faces and found that there was a significant difference in the visual scanning patterns of Han subjects’ recognition of their own-race faces and Tibetan faces. Specifically, the Han subjects gazed more on the eyes and mouth of Han faces than on the nose, gazed more on the eyes of Tibetan faces than on the nose and gazed more on the nose than on the mouth. This result is consistent with other studies [3,41] that believed that Han subjects used a scanning pattern from the face center to the surrounding area when recognizing Han and Tibetan faces. The differences between Tibetan and Han face scanning patterns are mainly caused by the differences in Tibetan and Han face configurations. For Han faces, the center is between the eyes, and the subjects paid more attention to the eyes in the center area. Moreover, studies have found that the mouth could reflect a large amount of emotional information [43,44], such as fear, surprise, happiness and anger. Subjects could obtain many emotional cues from the mouth, and they gazed more at the mouth; moreover, Han subjects gazed more at the eyes and mouth [3]. For Tibetan faces, the center is located at the upper bridge of the nose, and the subjects’ gaze on the nose and eyes near the center increased significantly. In addition, the nose of Tibetan faces is larger, which causes subjects to gaze more at it [18]. The eyes were regarded as the most important gaze area of the face by Han subjects, and the differences in their scanning patterns were mainly in the nose and mouth areas. The differences in the center of the face were the main reasons for the differences in the scanning patterns of the Tibetan and Han faces. The differences in visual scanning patterns of Han subjects processing Han and Tibetan faces may reveal the reason for the ORE, and the differences in scanning patterns may lead to differences in the accuracy of subjects’ recognition of their own and other ethnic faces.

### 4.2. The Regulatory Effect of Age on the ORE and Visual Scanning Patterns of Cross-Ethnic Face Recognition

The behavioral and eye movement data of the current study proved that aging was certainly good for reducing the ORE. As discussed above, the differences between Tibetan and Han face visual scanning patterns are mainly reflected in the fixations on the nose and mouth. Young Han subjects gazed more at the mouth of Han faces than on Tibetan faces and gazed more at the nose of Tibetan faces than on Han faces, while the fixation differences between the two ethnicities were reduced or eliminated in the elderly subjects. The eye movement data of the elderly subjects showed that there was no difference in the fixation counts between elderly subjects on the nose and mouth of the Tibetan faces, and the fixation counts of elderly subjects on the mouths of Han faces were greater than those on the nose. However, there was no difference in the fixation durations on the nose and mouth between Han and Tibetan faces. The fixation count represents the number of times a subject scans a certain facial feature, which could reflect the allocation of a subject’s attention resources to a certain extent, but the amount of attention a subject pays to a certain facial feature is mainly determined by the fixation duration (the fixation duration represents the amount of attention resources a subject allocates to a certain object). Therefore, the above data could at least show that the difference between elderly subjects’ gazes at the nose and mouth of own-race and Tibetan faces has decreased or disappeared, which is different from that of the young Han subjects.

We believe that this may be related to the increase in the processing of the vertical configuration of own- and other-ethnicity faces by elderly subjects. Elderly subjects are better at the vertical configuration processing of faces, mainly in the processing of the nose and mouth, than young subjects [20]. Elderly subjects use configuration processing when scanning the nose and mouth features of own- and other-ethnicity faces and regard the two areas as a whole. In addition, the perceptual aging of the elderly can also affect their visual scanning patterns of faces. Due to perceptual aging, elderly individuals have to focus on more vertical configuration information to complete face recognition tasks. Therefore, elderly subjects demonstrated little difference in scanning the nose and mouth features of own- and other-ethnicity faces. The reduction in the ORE of elderly subjects may be related to the increase in elderly subjects’ experience with faces of other races. With the development of regional transportation, the economy and the internet, cross-cultural communications between Han and Tibetan subjects have become increasingly frequent. In recent years, although Han subjects have had significantly less experience with Tibetan faces than Tibetan subjects, compared with young Han subjects, elderly Han subjects accumulate more experience with Tibetan faces. In addition, economic, trade and cultural exchanges between ethnic groups are becoming increasingly frequent, and elderly Han subjects have accumulated more experience with Tibetan faces than younger Han subjects. The increase in experience with Tibetan faces significantly increased the accuracy of recognition of Tibetan faces by elderly Han subjects, which led to a reduction in the ORE.

Overall, the elderly Han subjects’ ORE was lower than that of the young Han subjects. The visual scanning patterns of elderly subjects in recognizing faces of individuals that share their ethnicity and those of other ethnicities were relatively small, while the visual scanning patterns of young subjects in recognizing faces of individuals that share their ethnicity and those of other ethnicities were very significantly different. This may explain why elderly Han subjects have smaller OREs than young Han subjects.

## 5. Conclusions

Under the conditions of this experiment, compared with young Han subjects, elderly Han subjects had a lower ORE in recognizing Han and Tibetan faces, and the differences in scanning patterns of own- and other-ethnicity faces were lower.

## Figures and Tables

**Figure 1 behavsci-13-00675-f001:**
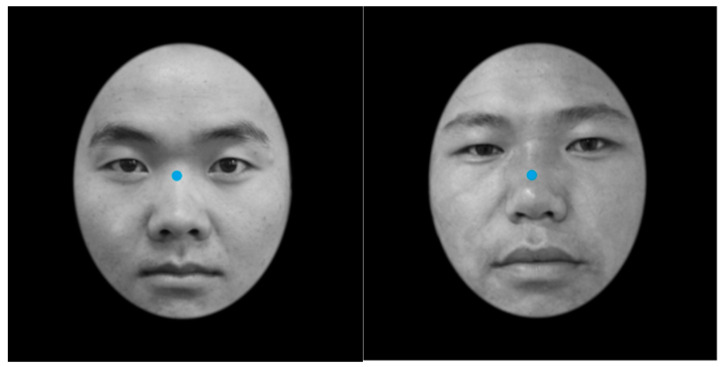
Face center of Han and Tibetan peoples. Note: On the left is a Han face, and on the right is a Tibetan face.

**Figure 2 behavsci-13-00675-f002:**
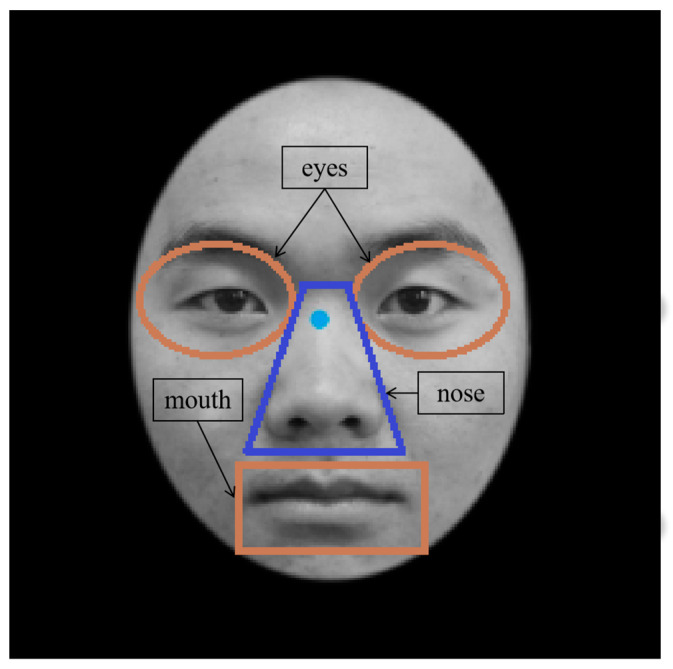
AOIs (the eyes, nose and mouth).

**Figure 3 behavsci-13-00675-f003:**
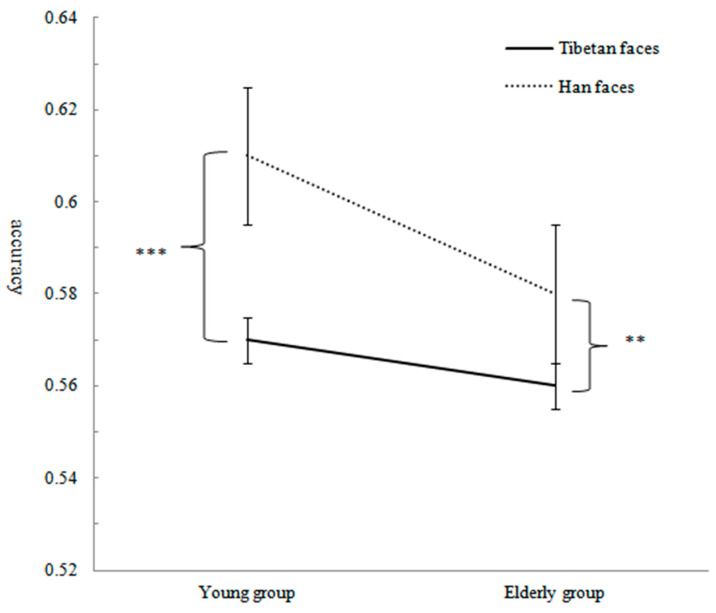
Accuracy and standard error for recognizing Tibetan and Han face eyes in different age groups. Note: ** *p* < 0.01; *** *p* < 0.001.

**Figure 4 behavsci-13-00675-f004:**
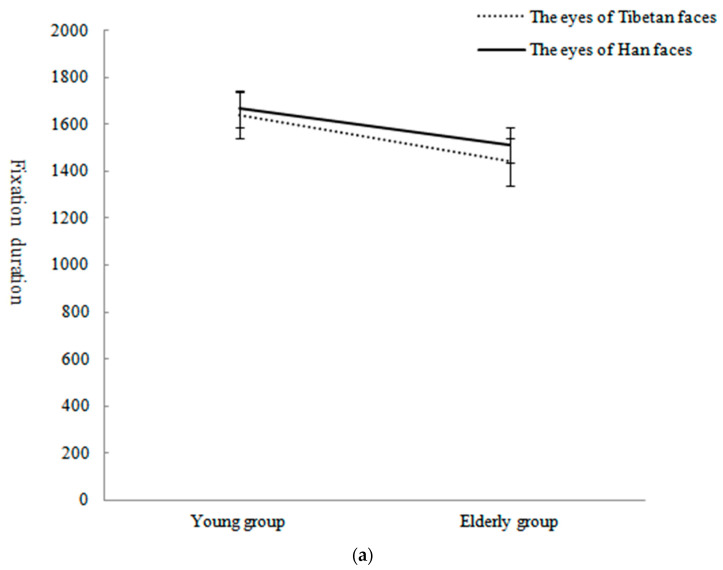
(**a**) Fixation duration and standard error for recognizing Tibetan and Han faces’ eyes in different age groups. (**b**) Fixation duration and standard error for recognizing Tibetan and Han faces’ nose in different age groups. (**c**) Fixation duration and standard error for recognizing Tibetan and Han faces’ mouth in different age groups. Note: *** *p* < 0.001.

**Figure 5 behavsci-13-00675-f005:**
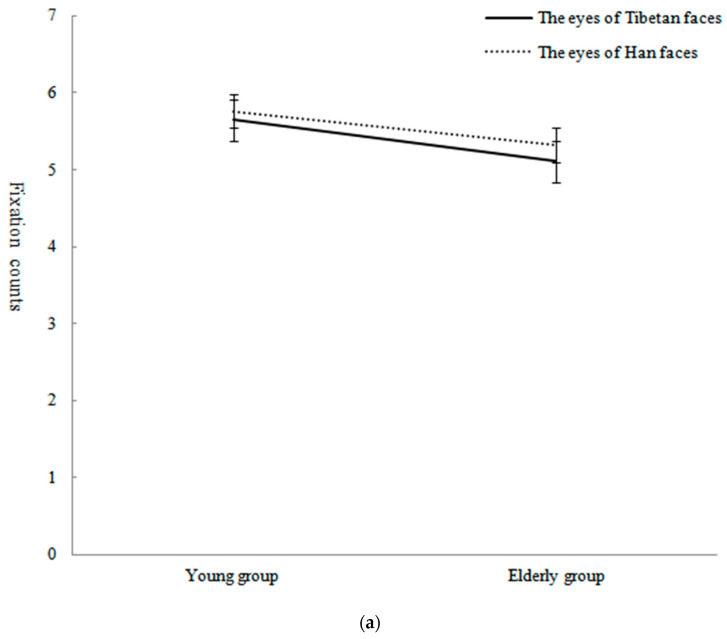
(**a**) Fixation counts and standard error for recognizing Tibetan and Han faces’ eyes in different age groups. (**b**) Fixation counts and standard error for recognizing Tibetan and Han faces’ nose in different age groups. (**c**) Fixation counts and standard error for recognizing Tibetan and Han faces’ mouth in different age groups. Note: *** *p* < 0.001.

**Table 1 behavsci-13-00675-t001:** Means and standard deviations (*M* ± *SD*) of the recognition accuracy of Tibetan and Han faces.

	Tibetan Face	Han Face
Young group	0.57 ± 0.16	0.61 ± 0.13
Elderly group	0.56 ± 0.15	0.58 ± 0.18

**Table 2 behavsci-13-00675-t002:** Means and standard deviations (*M* ± *SD*) of fixation durations and counts of AOIs for Tibetan and Han faces.

AOI	Fixation Duration		Fixation Count	
	Tibetan Faces	Han Faces	Tibetan Faces	Han Faces
Eyes	1540.28 ± 521.51	1589.43 ± 472.66	6.36 ± 2.00	5.52 ± 1.39
Nose	1222.62 ± 402.46	1007.95 ± 454.69	4.37 ± 1.15	3.45 ± 1.25
Mouth	952.91 ± 475.28	1380.56 ± 506.70	3.17 ± 1.49	5.00 ± 1.37

**Table 3 behavsci-13-00675-t003:** Means and standard deviations (*M* ± *SD*) of the fixation duration of different age groups for the AOIs of Tibetan and Han faces.

AOI	Young Group		Elderly Group	
	Tibetan Faces	Han Faces	Tibetan Faces	Han Faces
Eyes	1639.98 ± 525.24	1665.94 ± 487.42	1440.58 ± 506.80	1512.93 ± 454.09
Nose	1339.02 ± 408.51	903.00 ± 496.21	1106.22 ± 369.62	1112.90 ± 398.17
Mouth	865.12 ± 487.46	1488.68 ± 552.43	1040.70 ± 455.31	1272.42 ± 448.41

**Table 4 behavsci-13-00675-t004:** Means and standard deviations (*M* ± *SD*) of fixation counts (number of times) of subjects in different age groups for the AOIs of Tibetan and Han faces.

AOI	Young Group		Elderly Group	
	Tibetan Faces	Han Faces	Tibetan Faces	Han Faces
Eyes	5.65 ± 1.93	5.76 ± 2.11	5.11 ± 2.05	5.32 ± 1.78
Nose	4.58 ± 1.45	3.04 ± 1.20	4.20 ± 1.14	3.80 ± 1.20
Mouth	2.63 ± 1.39	5.07 ± 1.39	3.63 ± 1.42	4.94 ± 1.37

## Data Availability

The datasets generated and analyzed during the current study are available at https://pan.baidu.com/s/1KxuZv1EVnL1bGUFtuwIdjA?pwd=xrqz. accessed on 17 February 2023.

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
