# Peer review of "Age Weakens the Other-Race Effect among Han Subjects in Recognizing Own- and Other-Ethnicity Faces"

_behavsci, 2023, doi:10.3390/bs13080675_

Round 1

Reviewer 1 Report

The paper investigates the effect of age on the other race effect (both behaviourally and in eye movements) in Han subjects. The Authors found that elderly people have a reduced beahvioural ORE compared to younger participants and that elderly people tend to scan different faces in a similar way, while younger participants explore Han and Tibetan faces differently.

Overall, I think that the paper is investigating an interesting and important topic, which can help us better understand how we recognize faces. However, I think that some issues must be solved before the paper is suitable for publication.

One point that I found a little confusing throughout the paper is the relationship between the behavioural ORE and the scan path. I think that the Authors need to address this both in the intro and in discussing their results. For instance, some papers mentioned in the intro report a difference in how people from different races scan faces, regardless of the race of the face, and other studies found that faces of different races are explored differently regardless of the race of the viewer; I think that these ‘mixed’ findings raised the question of where the behavioural ORE is coming from, i.e., what is the relationship between eye movement and ORE? This is a relevant issue since one of the main goals of this paper is to compare the eye movements of young and old participants when looking at faces of different ethnicities.

Some more information is required for the method and result section. First, why were those ROIs used? I find that considering the eyes as a central region is too approximative. Why not also consider the difference between the left and the right eye? The Authors could also consider using a more data-driven analysis approach like the iMAP. Why were simple effect analyses not the post-hoc approach with some p-value correction? For instance, the behavioural ORE effect in the older group seems very small (about 2%), and it seems unlikely that this could be significant (notice that a non-significant ORE for the older group could further support the Authors’ hypothesis and match the eye-movements analyses). I would suggest the Authors include analyses on the mean fixation duration, too: indeed, fixation count and total duration often strongly correlate (see the very similar results in the paper), while the mean fixation durations might reveal something more about the processing. When reporting the triple interaction, comparisons between the faces across the same ROIs should be reported really understand the different scanning strategies for the two types of faces. All figures need to be edited to include values for the Y-axis and standard deviations (or standard error) information.

Minor points:

-       In the introduction, it might be worth reporting if any paper has found any difference in the scan path of young and old participants looking at faces regardless of the face race/ethnicity

-       Were the same pictures used in the learning and recognition phase? If so, this could be a problem for the behavioural results since participants might have used a picture-matching strategy more than actually recognizing the faces

-       Why were eye movements for the recognition phase where not analyzed?

Some parts would benefit from an English revision

Author Response

Dear Reviewer,

Thank you for the meticulous review and further thoughtful comments on our manuscript, which has benefited from these insightful suggestions. We have read all your comments carefully and have modified the manuscript accordingly. The modified content follows below.

To facilitate cross-referencing between the comments and corresponding text revisions, we have numbered each comment. We have also attached a clean version of the manuscript.

We hope that the revised (written in blue) version of this manuscript addresses your concerns. We will be glad to respond to any further questions or comments that may arise.

Referee: 1

Overall, I think that the paper is investigating an interesting and important topic, which can help us better understand how we recognize faces. However, I think that some issues must be solved before the paper is suitable for publication.

  1. One point that I found a little confusing throughout the paper is the relationship between the behavioural ORE and the scan path. I think that the Authors need to address this both in the intro and in discussing their results. For instance, some papers mentioned in the intro report a difference in how people from different races scan faces, regardless of the race of the face, and other studies found that faces of different races are explored differently regardless of the race of the viewer; I think that these ‘mixed’ findings raised the question of where the behavioural ORE is coming from, i.e., what is the relationship between eye movement and ORE? This is a relevant issue since one of the main goals of this paper is to compare the eye movements of young and old participants when looking at faces of different ethnicities.

Reply: Thank you for the kind suggestions. Researchers have used eye movement technology to research cross-race face recognition mainly to explore the reasons for ORE. We did not clarify this viewpoint in the previous version of the manuscript. To highlight the relationship between eye movement and ORE, we strengthened the discussion of this content in the introduction and discussion sections. The details are as follows:

In recent years, eye movement technology has been widely used to explore the differences in scanning patterns caused by the ORE because it can directly reflect the visual processing characteristics of faces [4, 5, 6]. Researchers hoped to explore the reasons for the ORE by comparing the cross-racial visual scanning patterns of faces. Studies of eye movement have found that subjects may use different scanning patterns when recognizing own- and other-race faces [7, 8, 9, 10]. Some studies found that the differences in processing strategies [4, 11] and face configurations were the main reasons for the differences in scanning patterns of other-race face recognition, and these differences may be the reason for the ORE [12, 13]. (See details, P. 2, 1 Introduction).

To further analyze the reasons for the ORE in Han subjects’ recognition of Tibetan and Han faces, we conducted a comparative analysis of the visual scanning patterns of Han subjects who scanned Han and Tibetan faces and found that there was a significant difference in the visual scanning patterns of Han subjects’ recognition of their own-race faces and Tibetan faces. (See details, P. 21, 4 Discussion)

The differences in visual scanning patterns of Han subjects processing Han and Tibetan faces may reveal the reason for the ORE, and the differences in scanning patterns may lead to differences in the accuracy of subjects’ recognition of their own and other ethnic faces. (See details, P. 22, 4 Discussion).

Overall, the elderly Han subjects' ORE was lower than that of the young Han subjects. The visual scanning patterns of elderly subjects in recognizing faces of individuals that share their ethnicity and those of other ethnicities were relatively small, while the visual scanning patterns of young subjects in recognizing faces of individuals that share their ethnicity and those of other ethnicities were very significantly different. This may explain why elderly Han subjects have smaller OREs than young Han subjects. (See details, P. 23, 4 Discussion).

  1. Some more information is required for the method and result section. First, why were those ROIs used? I find that considering the eyes as a central region is too approximative. Why not also consider the difference between the left and the right eye? The Authors could also consider using a more data-driven analysis approach like the iMAP. Why were simple effect analyses not the post-hoc approach with some p-value correction? For instance, the behavioural ORE effect in the older group seems very small (about 2%), and it seems unlikely that this could be significant (notice that a non-significant ORE for the older group could further support the Authors’ hypothesis and match the eye-movements analyses). I would suggest the Authors include analyses on the mean fixation duration, too: indeed, fixation count and total duration often strongly correlate (see the very similar results in the paper), while the mean fixation durations might reveal something more about the processing. When reporting the triple interaction, comparisons between the faces across the same ROIs should be reported really understand the different scanning strategies for the two types of faces. All figures need to be edited to include values for the Y-axis and standard deviations (or standard error) information.

Reply: Thank you for the comment. Due to the inclusion of multiple questions in this comment, we will answer each question separately to better respond to the concerns of the reviewers.

(1)Some more information is required for the method and result section. First, why were those ROIs used? I find that considering the eyes as a central region is too approximative. Why not also consider the difference between the left and the right eye? The Authors could also consider using a more data-driven analysis approach like the iMAP.

Reply: Thank you for the comment. This method of dividing regions of interest is mainly based on previous research, with one study specifically summarizing the main methods of dividing regions of interest [16], including the regions of interest for the eyes, nose, and mouth; left and right areas of interest of the face (left and right sides of the eyes, left and right sides of the nose, and left and right sides of the mouth); and upper and lower areas of interest of the face (upper and lower parts of the face). This study used the most commonly used method of dividing facial interest regions, which is often used to explore facial visual scanning patterns. The left and right facial interest regions divided the eyes into two interest regions, mainly used to explore the lateralization trend of facial visual processing[4]. In addition, dividing the eyes into the same area of interest can better reflect the configuration processing between the eyes (such as the distance between the eyes), which is an indispensable and important process in face recognition. Finally, when recognizing faces, people often focus on both eyes simultaneously, and previous studies have found that individuals have no difference in their gaze toward the left and right eyes. Based on the above reasons, we did not divide the eyes into two interest regions.

Using data-driven analysis methods enables fully mining data such as AOI maps, which can more directly reflect the individual's fixation patterns toward different areas of interest. Notably, the AOI Map has a certain degree of consistency with fixation counts and duration, and compared to the latter, the AOI Map may not be able to more accurately reflect the differences in fixation between regions of interest. Therefore, we used interaction figures (P. 14-15, Figure 4a-4c; P. 18-19, Figure 5a-4c) to reflect the differences in scanning patterns for faces of different races and ages. In these figures, we use error bars and asterisks to accurately reflect the fixation of different AOIs.

Overall, the method of dividing regions of interest and analyzing eye movement data used in this study mainly referred to previous studies. Based on these studies, we selected representative data (fixation counts and duration) to reflect the differences in scanning patterns between ethnic groups.

(2) Why were simple effect analyses not the post-hoc approach with some p-value correction? For instance, the behavioural ORE effect in the older group seems very small (about 2%), and it seems unlikely that this could be significant (notice that a non-significant ORE for the older group could further support the Authors’ hypothesis and match the eye-movements analyses).

Reply: Thank you for the comment. Based on your suggestion, we have reanalyzed all the results and used Boffi correction (see details, P. 11, 3.2. Eye movement data). The results have not changed much from the original results. We have updated the results in the manuscript and uploaded the experimental results to the system with the attachment. Notably, the ORE effect of elderly participants did not disappear after using Boffi correction. We analyzed the reasons and found that although the behavioral ORE effect in the older group faces was very small (approximately 2%), the confidence intervals for the accuracy of elderly participants in recognizing different ethnic faces were (Han: .57-.60) and (Tibetan: .55-.58), respectively. This indicates that although the advantages of elderly participants in recognizing faces from different ethnic groups are small, they are stable.

(3)I would suggest the Authors include analyses on the mean fixation duration, too: indeed, fixation count and total duration often strongly correlate (see the very similar results in the paper), while the mean fixation durations might reveal something more about the processing.

Reply: Thank you for the comment. Based on your suggestion, we analyzed the mean fixation duration and found only one significant interaction (the interaction between AOI and face ethnicity was significant, F(2, 150)=6.02, p<.01, ηp²=.07. Through simple effect analysis, it was found that the subjects' mean fixation for the mouths of Tibetan faces was significantly greater than that for Han faces, F(1, 75)=13.42, p<.001, ηp²=.15, the mean fixation for the eyes (F(1, 75)=0.013, p=.091, ηp²=.013) and nose (F(1, 75)=3.28, p=.07, ηp²=.04) for Han and Tibetan faces was not significantly different, and no other significant main or interaction effects were found. We believe that compared to the total fixation duration, the mean fixation duration may narrow the differences between groups, resulting in insignificant differences. Due to the lack of new findings in the analysis of the mean fixation duration, this result was not presented in the results section. We have uploaded the data analysis results in the attachment (file name: mean fixation duration).

(4)When reporting the triple interaction, comparisons between the faces across the same ROIs should be reported really understand the different scanning strategies for the two types of faces.

Reply: Thank you for the comment. We have added the corresponding analysis to the results section, and the detailed content is as follows:

The fixation durations for the mouths of Han faces were significantly greater than those for Tibetan faces, F(1, 75)=22.03, p<.001, ηp²=.23. The fixation durations for the nose of Tibetan faces were significantly greater than those for Han faces, F(1, 75)=7.89, p<.01, ηp²=.10, and the fixation durations for the eyes of Han and Tibetan faces were not significantly different, F(1, 75)=1.02, p=.32, ηp²=.01. (See details, P. 12, 3 Results).

When the young subjects recognized Han and Tibetan faces, the fixation durations for the mouth of Han faces were significantly greater than those for Tibetan faces, F(1, 75)=21.47, p<.001, ηp²=.22; the fixation durations for the nose of Tibetan faces were significantly greater than those for Han faces, F(1, 75)=14.93, p<.001, ηp²=.17; and the fixation durations for the eyes of Han and Tibetan faces were not significantly different, F(1, 75)=.13, p=.72, ηp²=.002. When the elderly subjects recognized Han and Tibetan faces, the fixation durations for the mouth (F(1, 75)=15.22, p<.001, ηp²=.17) of Han faces were significantly greater than those for Tibetan faces, and the fixation durations for the eyes (F(1, 75)=1.21, p=.27, ηp²=.016), nose (F(1, 75)=.004, p=.95, ηp²=.000) and mouth (F(1, 75)=3.56, p=.06, ηp²=.045) of Han and Tibetan faces were not significantly different. (See details, P. 13, 3 Results).

The fixation counts for the mouths (F(1, 75)=21.94, p<.001, ηp²=.23) of Han faces were significantly greater than those for Tibetan faces, F(1, 75)=56.76, p<.001, ηp²=.43; the fixation counts for the noses of Tibetan faces were significantly greater than those for Han faces, F(1, 75)=21.94, p<.001, ηp²=.23; and the fixation counts for the eyes of Han and Tibetan faces were not significantly different, F(1, 75)=.71, p=.40, ηp²=.01. (See details, P. 16, 3 Results).

When the young subjects recognized Han and Tibetan faces, the fixation counts for the mouths of Han faces were significantly greater than those for Tibetan faces, F(1, 75)=44.08, p<.001, ηp²=.37; the fixation counts for the noses of Tibetan faces were significantly greater than those for Han faces, F(1, 75)=25.45, p<.001, ηp²=.25; and the fixation counts for the eyes of Han and Tibetan faces were not significantly different, F(1, 75)=.14, p=.72, ηp²=.002. When the elderly subjects recognized Han and Tibetan faces, the fixation counts for the mouths (F(1, 75)=15.22, p<.001, ηp²=.17) of Han faces were significantly greater than those for Tibetan faces, and the fixation counts for the eyes (F(1, 75)=.70, p=.41, ηp²=.009) and noses (F(1, 75)=2.10, p=.16, ηp²=.03) of Han and Tibetan faces were not significantly different. (See details, P. 17, 3 Results).

(5)All figures need to be edited to include values for the Y-axis and standard deviations (or standard error) information.

Reply: Thank you for the comment. Based on your suggestion, we have made modifications to these figures. (See details, P. 11, Figure 3; P. 14-15, Figure 4a-4c; P. 18-19, Figure 5a-4c). The details are as follows(For details, see the attachment):

Note: *P<0.05; **P<0.01; ***P<0.001 (the same below).

Figure 4a. Fixation duration and standard error for recognizing Tibetan and Han faces eyes in different age groups.

Figure 4b. Fixation duration and standard error for recognizing Tibetan and Han faces nose in different age groups.

Figure 4c. Fixation duration and standard error for recognizing Tibetan and Han faces mouth in different age groups.

Figure 5a. Fixation counts and standard error for recognizing Tibetan and Han faces eyes in different age groups.

Figure 5b. Fixation counts and standard error for recognizing Tibetan and Han faces nose in different age groups.

Figure 5c. Fixation counts and standard error for recognizing Tibetan and Han faces mouth in different age groups.

Minor points:

  1. - In the introduction, it might be worth reporting if any paper has found any difference in the scan path of young and old participants looking at faces regardless of the face race/ethnicity

Reply: Thank you for the comment. We have added eye movement research on the facial scanning patterns of young and elderly people in the introduction section, as follows:

Existing studies have mainly focused on the differences in scanning patterns between early adulthood and elderly adulthood when individuals recognize faces of individuals from their own ethnic group [40]. For example, compared to young people, elderly subjects tend to adopt a holistic mode, in which most eye gaze is concentrated in the center of the face, especially elderly people with lower cognitive states [42]. For facial features, both young and elderly people consider the eyes and nose as the main focus areas [40], whereas elderly people tend to pay more attention to their mouths than young people [43]. However, focusing on only how young and elderly people process the faces of individuals from their own ethnic groups is not sufficient, and thus far, whether the ORE exists in elderly adulthood is unclear. Therefore, this study selected Tibetan and Han faces as materials, took young and elderly Han people as subjects, and used ethology and eye movement techniques to explore the impact of aging on the ORE and other-race face scanning patterns of Han subjects. (See details, P. 5, 1 Introduction).

5.-       Why were eye movements for the recognition phase where not analyzed?

Reply: Thank you for the comment. In this study, no eye movement data were recorded during the testing phase. During the testing phase, participants were asked to press buttons to complete the recognition task, which may have affected the collection of eye movement data. Therefore, we did not collect eye movement data from participants during the testing phase. In addition, research has found a high similarity in eye movement data between the learning and recognition phases (Henderson, Williams, &Falk, 2005), and most studies collect eye movement data only from the learning or recognition phase (Cangöz et al., 2013; Chan et al., 2018; Ma et al., 2022), which is another reason this study collected eye movement data only from the learning stage.

We hope the above modifications can effectively address your concerns about the manuscript.

Reviewer 2 Report

The manuscript presents an investigation on the other race effect (ORE) in different age groups of Han (Chinese ethnicity) individuals, utilizing behavioral and eye-tracking methods. Overall, the manuscript is well-written and well-conducted, but some minor changes can be made to enhance its quality:

Abstract:

- In the sentence "The behavioral results showed that the ORE of elderly Han subjects was significantly lower than that of young Han subjects," it should be specified in which task the elderly Han performed lower than the young Han subjects.

Introduction:

- It is important to clarify which faces represent the Han and Tibetan ethnicities in Figure 1, either in the figure itself or in its legend.

Methods:

- Did all participants live in communities in contact with members of both ethnicities?

- Please provide the acquisition rate of eye-tracking recordings.

- In the manuscript, both recognition performance and recognition accuracy are mentioned. Are they the same? If not, please explain how both are calculated.

Results:

- The title of Table 1 mentions recognition accuracy and reaction time, but it is not clear what the numbers in the table represent.

- For all tables, please use asterisks to indicate statistical significance for differences, when applicable.

Discussion:

- Throughout the manuscript, there is no mention of visual changes related to aging, such as the decrease in luminance contrast sensitivity, color vision changes, consequences of cataracts, and other impairments of visual mechanisms. Including this information would be valuable to the Discussion section.

By addressing these minor points, the manuscript will be further improved and provide a more comprehensive and accurate presentation of the research findings.

Author Response

Dear Reviewer,

Thank you for the meticulous review and further thoughtful comments on our manuscript, which has benefited from these insightful suggestions. We have read all your comments carefully and have modified the manuscript accordingly. The modified content follows below.

To facilitate cross-referencing between the comments and corresponding text revisions, we have numbered each comment. We have also attached a clean version of the manuscript.

We hope that the revised (written in blue) version of this manuscript addresses your concerns. We will be glad to respond to any further questions or comments that may arise.

Referee: 2

Abstract:

1.- In the sentence "The behavioral results showed that the ORE of elderly Han subjects was significantly lower than that of young Han subjects," it should be specified in which task the elderly Han performed lower than the young Han subjects.

Reply: Thank you for the comment. Based on your suggestion, we have added this content to the abstract as follows:

All participants were asked to complete the study recognition task, and the behavioral results showed that the ORE of elderly Han subjects was significantly lower than that of young Han subjects. (See details, P. 5, Abstract)

 Introduction:

  1. - It is important to clarify which faces represent the Han and Tibetan ethnicities in Figure 1, either in the figure itself or in its legend.

Reply: Thank you for the comment. Based on your suggestion, we have added a "note" below Figure 1 to indicate which faces represent the Han and Tibetan ethnicities, as follows:

Note:On the left is a Han face, and on the right is a Tibetan face. (See details, P. 7, Figure 1).

Methods:

  1. - Did all participants live in communities in contact with members of both ethnicities?

Reply: Thank you for the comment. All participants came from concentrated areas of Han, and Han participants had no chance to come into contact with Tibetans in their daily lives. We also provided explanations in the subject section (see details, P. 2, 2.1). Participants).

  1. - Please provide the acquisition rate of eye-tracking recordings.

Reply: Thank you for the comment. All subjects' eye movement data in this study were collected, and we provided explanations in the data analysis section. As follows:

A total of 77 effective eye movement data points were collected (35 young subjects, 42 elderly subjects). (See details, P. 6, 2.5. Research design and data analysis).

  1. - In the manuscript, both recognition performance and recognition accuracy are mentioned. Are they the same? If not, please explain how both are calculated.

Reply: Thank you for the comment. Both recognition performance and recognition accuracy represent accuracy, and we have uniformly replaced them with recognition accuracy.

Results:

  1. - The title of Table 1 mentions recognition accuracy and reaction time, but it is not clear what the numbers in the table represent.

Reply: Thank you for the comment. We apologize for this error. Due to some elderly participants being unable to proficiently complete key pressing tasks (the test task was completed through oral reports from elderly participants, and the experimenter recorded it), their reaction times could not be accurately recorded. Therefore, we did not record the participants' reaction times, and we also deleted this statement in the results section.

  1. - For all tables, please use asterisks to indicate statistical significance for differences, when applicable.

Reply: Thank you for the comment. We attempted to add asterisks to indicate statistical significance for differences, but this would lead to some confusion when reading the table. To avoid such confusion and more clearly indicate statistical significance, we marked the statistical significance in the interaction graph as shown in Figure 2-4. (See details, P. 11, Figure 3; P. 14-15, Figure 4a-4c; P. 18-19, Figure 5a-4c).

Discussion:

  1. - Throughout the manuscript, there is no mention of visual changes related to aging, such as the decrease in luminance contrast sensitivity, color vision changes, consequences of cataracts, and other impairments of visual mechanisms. Including this information would be valuable to the Discussion section.

Reply: Thank you for the kind suggestion. According to your suggestion, we added the possible impact of perceptual aging on facial visual scanning patterns in elderly people. The details are as follows:

Elderly subjects are better at the vertical configuration processing of faces, mainly in the processing of the nose and mouth, than young subjects [20]. Elderly subjects use configuration processing when scanning the nose and mouth features of own- and other-ethnicity faces and regard the two areas as a whole. In addition, the perceptual aging of the elderly can also affect their visual scanning patterns of faces. Due to perceptual aging, elderly individuals have to focus on more vertical configuration information to complete face recognition tasks. (See details, P. 22-23, 4 Discussion)

By addressing these minor points, the manuscript will be further improved and provide a more comprehensive and accurate presentation of the research findings.

We hope the above modifications can effectively address your concerns about the manuscript.

Round 2

Reviewer 1 Report

The Authors have replied satisfactorily to all my comments. I feel that the edited version of the manuscript is acceptable for publication. Thanks.

Reviewer 2 Report

I have no more issues to be addressed.